# CONCEPT DENOISING SCORE MATCHING FOR RESPONSIBLE TEXT-TO-IMAGE GENERATION

**Silpa Vadakkeeveetil Sreelatha [1], Sauradip Nag [2], Serge Belongie [3],**
**Muhammad Awais [1], Anjan Dutta [1]**
[1] University of Surrey  [2] Simon Fraser University  [3] University of Copenhagen

## ABSTRACT

Diffusion models excel at generating diverse, high-quality images, but they also risk producing unfair and harmful content. Existing methods that update text embeddings or model weights either fail to address biases within diffusion models or are computationally expensive. We tackle responsible (fair and safe) text-to-image (T2I) generation in diffusion models as an interpretable concept discovery problem, introducing **Co**ncept **D**enoising **S**core **Ma**tching (CoDSMa) – a novel objective that learns responsible concept representations in the bottleneck feature activation (*h-space*). Our approach builds on the observation that, at any timestep, aligning the neutral prompt with the target prompt directs the predicted score of denoised latent towards the target concept. We empirically demonstrate that our method enables responsible T2I generation by addressing two key challenges: mitigating gender and racial biases (fairness) and eliminating harmful content (safety). Our approach reduces biased and harmful generation by nearly 50% compared to state-of-the-art methods. Remarkably, it outperforms other techniques in debiasing gender and racial attributes without requiring profession-specific data. Furthermore, it successfully filters inappropriate content, such as depictions of illegal activities or harassment, without training on such data.

## 1 INTRODUCTION

The rise of text-to-image diffusion models (T2I), such as Stable Diffusion, has significantly impacted content creation and visual communication by enabling high-quality visuals from simple text prompts (Rombach et al., 2022; Podell et al., 2024). However, these models risk reinforcing stereotypes or generating harmful content, leading to societal consequences (Luccioni et al., 2023; Perera & Patel, 2023; Rando et al., 2022; Schramowski et al., 2023). Ensuring a responsible workflow that prioritizes fair and safe generation is critical to reducing these risks.

In this work, we address responsible T2I generation through interpretable representation learning within the feature activations of the bottleneck layer in diffusion models, specifically the *h-space*, as introduced in Li et al. (2024). We define 'responsible concepts' as attributes related to both fairness and safety. Unlike Li et al. (2024), which identifies concepts in the $h$-space using generated images – a computationally expensive process – we propose an alternative approach that leverages denoised latent representations. Inspired by visualizations from Katzir et al. (2024) on the denoising score components in diffusion models, we explore the following: given the denoising latent for a neutral prompt at timestep $t$ (the neutral denoising latent), how does modifying the neutral prompt to a target prompt affect the denoising score? Our findings show that at any timestep, the target prompt directs the predicted denoising score (target score) to steer neutral denoised latents toward the target concept. We use these target scores to learn concept representations in diffusion models. Further details on our setup, observations, and visualizations are in section 3.2.

Building on our empirical observations regarding the role of the target score, we introduce Concept Denoising Score Matching (CoDSMa), a novel score-matching objective designed to learn responsible concept representations in the $h$-space. Previous work Kwon et al. (2023) has demonstrated that semantic latent manipulation of images can be achieved through linear transformations in the $h$-space, making it a strong candidate for the concept representation learning in diffusion models. Given a neutral prompt and a responsible concept (target concept), our goal is to learn a vector,

referred to as the $c$-vector, which can be linearly added to the $h$-space to introduce interpretable variations in the generated images, corresponding to a responsible concept. We achieve this by introducing an objective that aligns the denoising score with the target score. Additionally, we demonstrate that updating the $c$-vector in the direction of the gradient of CoDSMa steers the image generation toward the target concept.

We empirically demonstrate the effectiveness of our approach for responsible T2I image generation, focusing on fairness and safe generation. Our method successfully mitigates gender and racial biases in profession-related images, without requiring training on profession-specific data, outperforming existing methods. We provide both quantitative and qualitative analyses showing that our objective effectively reduces the generation of inappropriate content.

Our work makes the following key contributions. (1) Study of the intermediate denoising score reveals that modifying the neutral prompt to the target prompt at any timestep guides the predicted denoising score to direct the neutral denoised latent towards the target concept. (2) Leveraging insights from our empirical observations, we propose CoDSMa, a novel concept score distillation technique for uncovering responsible concepts within the $h$-space of diffusion models. (3) Through extensive quantitative and qualitative analysis, we demonstrate that CoDSMa enhances the fairness and safety of T2I diffusion models, reducing unfair and inappropriate image generation by approximately 50% compared to existing counterparts.

## 2   BACKGROUND

**Responsible Generation using Diffusion Models:** Recent work has seen a surge in methods to mitigate biased and inappropriate content generation in Stable Diffusion models. Some approaches modify input prompts by removing problematic words (Schramowski et al., 2023; Ni et al., 2023), while others use prompt-tuning techniques (Kim et al., 2023) or learn projection embeddings on prompt representations (Chuang et al., 2023) to filter out undesirable content. However, these methods primarily focus on text-based features and overlook the latent features that propagate through the diffusion process. The authors of Gandikota et al. (2023); Shen et al. (2024) address this by fine-tuning model weights to suppress harmful content generation, but these approaches can be computationally expensive. Alternative methods like those in Schramowski et al. (2023) use classifier-free guidance to steer image generation away from undesirable content without additional training. Approaches proposed in Gandikota et al. (2024); Chuang et al. (2023) offer efficient, closed-form solutions for embedding matrices to ensure responsible generation, though they lack adaptability and fine generation control. Recent works (Parihar et al., 2024; Li et al., 2024) modify the bottleneck activations of diffusion models to ensure appropriate content generation. Similarly, our method utilizes bottleneck activations but introduces a novel objective based on intermediate denoised latents, enabling the discovery of responsible directions in the latent space of diffusion models.

**Concept Discovery in $h$-space:** Kwon et al. (2023) were the first to identify the bottleneck layer of U-Net ($h$-space) as the semantic latent space, providing evidence that manipulations within the $h$-space result in semantically meaningful and interpretable changes in the generated images. Their method leverages CLIP classifiers to learn disentangled representations in the $h$-space, but this comes at a high computational cost. In contrast, approaches like Haas et al. (2024) apply PCA decomposition in the $h$-space, while Park et al. (2023) derive a local latent basis within the space by utilizing the pullback metric associated with features to discover interpretable directions in an unsupervised manner. Li et al. (2024) identifies interpretable directions for a given target concept by using Stable Diffusion-generated images that align with the target concept. Our approach differs from Li et al. (2024) by identifying concepts through the intermediate denoised latent space representations in diffusion models, enabling a more efficient and precise manipulation of underlying features, rather than relying on the generated images themselves, which can be computationally expensive and may obscure concept representations.

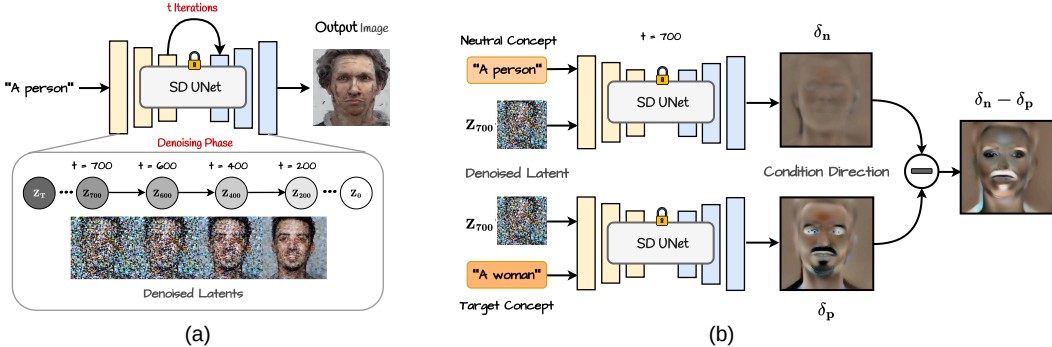

Figure 1: Visualization of condition directions at timestep $t = 700$.

## 3 METHODOLOGY

### 3.1 PROBLEM DEFINITION AND FORMULATION

This section presents a novel approach to enhancing fairness and safety in T2I diffusion models. We reframe the problem as identifying responsible concept representations within the diffusion models which enables unbiased and safe generation. We begin with a neutral prompt $y$ (*e.g.*, "a person") and a target prompt $y_p$ (*e.g.*, "a woman") representing a responsible concept. Our objective is to identify a direction, termed as $c$-vector within the $h$-space of a pre-trained T2I diffusion models. The $c$-vector, when applied as a linear transformation to the representations of the neutral prompt, induces semantically meaningful changes in the generation, aligning to the target concept.

In section 3.2, we present visualizations of the intermediate score estimates at various time steps, conditioned on denoised neutral latents and target prompts. Building on this observation, we introduce CoDSMa, which discovers target concept representations in the $c$-vector, as described in section 3.3. We also demonstrate how these vectors can directly improve fairness and safety in diffusion models without additional training, as discussed in section 3.4, with an illustration in fig. 2.

### 3.2 SCORE VISUALIZATIONS

This section presents visualizations and key observations of the intermediate score, which serve as the foundation for our proposed approach. Katzir et al. (2024) introduces an insightful decomposition of score components into interpretable elements. Their work visualizes the condition direction $\delta_C = \epsilon_\theta(\boldsymbol{z}_t; y, t) - \epsilon_\theta(\boldsymbol{z}_t; y = \varnothing, t)$, showing that it is interpretable and consistently aligns with the conditioning $y$ across various timesteps $t$ in the diffusion process. Inspired by Katzir et al. (2024), we conduct a similar analysis of the condition direction $\delta_C$ to explore how modifying the prompt conditioned on the neutral denoised latents at intermediate timesteps affects the predicted denoising score. Specifically, we use $y = $ "a person" and $y_p = $ "a woman" for this analysis. We begin by generating the denoised latent $\boldsymbol{z_t}$ at various timesteps $t$ for the neutral prompt $y$, as shown in fig. 1(a). We now consider $z_t$ at $t = 700$ as illustrated in fig. 1(b).

Next, we obtain U-Net predictions at $t = 700$ for two scenarios: (1) $\epsilon_\theta(\boldsymbol{z}_t; y, t)$, where prompt $y$ along with $\boldsymbol{z_t}$ is given as the input, (2) $\epsilon_\theta(\boldsymbol{z}_t; y_p, t)$, where target prompt $y_p$ along with $\boldsymbol{z_t}$ is given as the input. Let the corresponding condition directions be $\delta_n$ and $\delta_p$ respectively. We then visualize $\delta_n$, $\delta_p$, and their difference $\delta_n - \delta_p$ at $t = 700$, as illustrated in fig. 1 (b).

We observe that $\delta_n$ in fig. 1(b) aligns with the conditioning variable $y$ during the diffusion process, reinforcing the findings of Katzir et al. (2024). However, when the prompt $y_p$ is provided alongside the denoised latent $\boldsymbol{z_t}$ to generate U-Net predictions at timestep $t = 700$, the conditioning direction $\delta_p$ begins to emphasize attributes unrelated to the target concept. For instance, in fig. 1(b), features such as a beard and mustache become more prominent in the visualization of $\delta_p$. This shift occurs because the U-Net predicts the noise that needs to be removed from the neutral denoised latent $\boldsymbol{z_t}$ to guide it toward the target concept $y_p$. These observations empirically demonstrate that the target score $\epsilon_\theta(\boldsymbol{z}_t; y_p, t)$ steers the neutral denoised latent representations toward the target concept while preserving the original neutral concept. The difference term $\delta_n - \delta_p$ in fig. 1(b) further supports

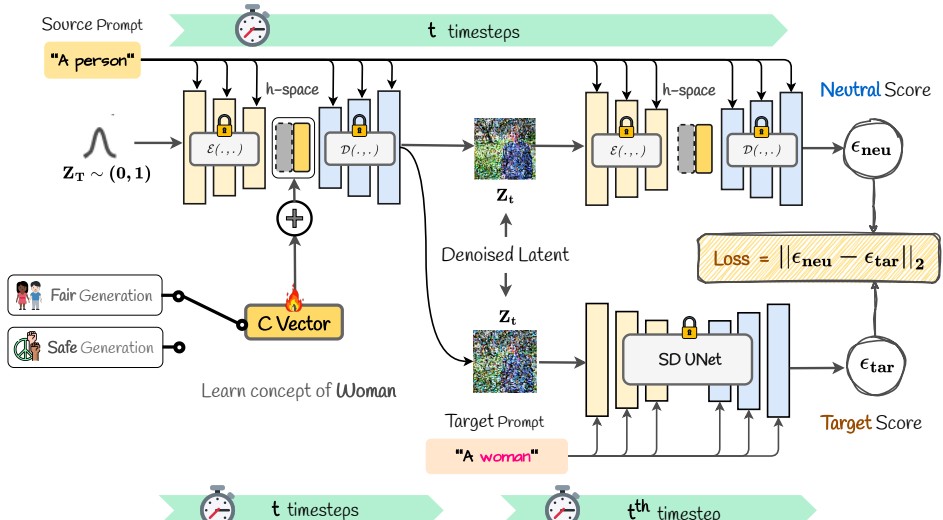

Figure 2: CoDSMa uses pretrained, frozen SD to guide generation toward fair, safe concepts. Reverse diffusion to timestep $t$ with a $c$-vector and "a person" prompt yields latent $z_t$. Forward diffusion with "a person" and $z_t$ predicts neutral score. Forward diffusion with "a woman" and $z_t$ predicts target score. CoDSMa aligns the scores which in turn updates the $c$-vector. SD weights are shared; no backpropagation through the reverse process.

this as it increasingly reflects the target concept. Based on these findings, we propose leveraging the target score $\epsilon_\theta(\boldsymbol{z}_t; y_p, t)$ to identify representations corresponding to the target concept $y_p$.

### 3.3 CONCEPT DENOISING SCORE MATCHING

Our objective is to discover interpretable representations in the $h$-space corresponding to the target concept $y_p$. Since the $h$-space (Li et al., 2024) of U-Net is designed to represent compressed and abstracted semantic features of the data (e.g, object shapes, structure, textures), we aim to learn a concept vector $c \in \mathbb{R}^D$, where $D$ is the dimension of the $h$-space. The $c$-vector is randomly initialized at the beginning of the training.

We start by decomposing the pretrained, frozen U-Net parameters of the diffusion model, $\theta$, into $\theta = \{\theta_1, \theta_2\}$, where $\theta_1$ denotes the frozen parameters of the U-Net encoder (denoted by $\mathcal{E}(.)$) including the bottleneck layers ($h$-space), and $\theta_2$ represents the parameters of U-Net decoder (denoted by $\mathcal{D}(.)$). Then, the score prediction function can be defined as follows:

$$\epsilon_\theta(\boldsymbol{z}_t; y, t) = \mathcal{D}_{\theta_2}(\mathcal{E}_{\theta_1}(\boldsymbol{z}_t; y, t); y, t) \tag{1}$$

If we substitute $\boldsymbol{h} = \mathcal{E}_{\theta_1}(\boldsymbol{z}_t; y, t)$ in eq. (1), where $\boldsymbol{h}$ represents the output of the middle bottleneck layer, the score prediction function simplifies to $\epsilon_\theta(\boldsymbol{z}_t; y, t) = \mathcal{D}_{\theta_2}(\boldsymbol{h}; y, t)$. The gradients of $\mathcal{L}_{\text{diff}}$ with respect to $\boldsymbol{h}$ is then given by:

$$\nabla_{\boldsymbol{h}}\mathcal{L}_{\text{diff}} = (\epsilon_\theta(\boldsymbol{z}_t; y, t) - \epsilon)\frac{\partial\epsilon_\theta(\boldsymbol{z}_t; y, t)}{\partial\mathcal{D}}\frac{\partial\mathcal{D}}{\partial\boldsymbol{h}} = \underbrace{(\mathcal{D}_{\theta_2}(\boldsymbol{h}; y, t) - \epsilon)}_{\text{Noise Residual}}\underbrace{\frac{\partial\mathcal{D}_{\theta_2}}{\partial\boldsymbol{h}}}_{\text{UNet Decoder Jacobian}} \tag{2}$$

In practice, the U-Net Jacobian term is expensive to compute (requires backpropagating through the diffusion U-Net), Since our aim is to learn representations in the $h$-space, *the gradient only flows through the U-Net decoder to the $h$-space*, which is comparatively less expensive to compute. It simply acts like an efficient, frozen critic that outputs $h$-space vectors. To facilitate the learning of concept representations in the $h$-space, we introduce learnable $c$-vector, similar to the approach in Li et al. (2024), that can be linearly added to the $h$-space vectors at each decoding timestep. Notably, our approach learns a single $c$-vector representing a concept that captures aggregate information across timesteps. The gradients of $\mathcal{L}_{\text{diff}}$ with respect to $c$ can be written as:

$$\nabla_{\boldsymbol{c}}\mathcal{L}_{\text{diff}} = (\mathcal{D}_{\theta_2}(\boldsymbol{h} + \boldsymbol{c}; y, t) - \epsilon)\frac{\partial\mathcal{D}_{\theta_2}}{\partial\boldsymbol{c}} \tag{3}$$

The above equation represents the optimization of the $c$-vector with respect to the standard diffusion loss. To facilitate concept discovery in the learnable $c$-vector, we now introduce **CoDSMa**, a score-matching objective. As outlined in section 3.2, the target score $\epsilon_\theta(z_t, y_p, t)$ effectively encodes the information necessary to uncover target concept representations. Our score-matching objective aligns the denoising scores with these target scores, which are then used to optimize the $c$-vector.

Since we utilize U-Net in both the presence of learnable $c$-vector and otherwise during the training, for notational clarity, we denote the denoising score as $\epsilon_\theta(z; h + c, y, t)$ to represent the presence of learnable $c$-vector which goes as the input to $\mathcal{D}$. We first randomly sample a timestep $t$ and obtain the denoised latent $z_t$ corresponding to the neutral prompt $y$ through the reverse process utilizing the learnable U-Net, where the denoising score is denoted by $\epsilon_\theta(z_t, h + c, y, t)$. We then provide the target prompt $y_p$ and $z_t$ to pretrained U-Net without $c$-vector to obtain the target score, which is given by $\epsilon_\theta(z_t, h, y_p, t)$. These scores are represented as $\epsilon_{\text{neu}}$ and $\epsilon_{\text{tar}}$ respectively in fig. 2. Then, the CoDSMa objective is defined as:

$$\mathcal{L}_{\text{CoDSMa}} = \parallel \epsilon_\theta(z_t; h + c, y, t) - \epsilon_\theta(z_t; h, y_p, t) \parallel_2 \tag{4}$$

We build on the observation in section 3.2 that $\epsilon_\theta(z_t, h, y_p, t)$ guides the denoised latent $z_t$ toward the target concept, which we aim to capture in the concept vector $c$ via our matching loss. In practice, we avoid backpropagating through the reverse process that outputs $z_t$ during $c$-vector learning due to high computational cost. Equation (4) can be expressed in terms of the U-Net decoder $\mathcal{D}$ as shown in eq. (1), given by:

$$\mathcal{L}_{\text{CoDSMa}} = \parallel \mathcal{D}_{\theta_2}(h + c; y, t) - \mathcal{D}_{\theta_2}(h; y_p, t) \parallel_2 \tag{5}$$

The gradient of $\mathcal{L}_{\text{CoDSMa}}$ w.r.t the $c$-vector can be written as:

$$\nabla_c \mathcal{L}_{\text{CoDSMa}} = (\mathcal{D}_{\theta_2}(h + c; y, t) - \mathcal{D}_{\theta_2}(h; y_p, t)) \frac{\partial \mathcal{D}_{\theta_2}}{\partial c} \tag{6}$$

By adding and subtracting the term $\epsilon$ in eq. (6), we can represent $\mathcal{L}_{\text{CoDSMa}}$ as a difference between gradients of two diffusion denoising score matching functions in eq. (3).

$$\nabla_c \mathcal{L}_{\text{CoDSMa}} = \nabla_c \mathcal{L}_{\text{diff}}(h + c, y, t) - \nabla_c \mathcal{L}_{\text{diff}}(h, y_p, t) \tag{7}$$

The overall gradient $\nabla_c \mathcal{L}_{\text{CoDSMa}}$ points in the direction that minimizes the difference between the two gradients $\nabla_c \mathcal{L}_{\text{diff}}(h + c, y, t)$ and $\nabla_c \mathcal{L}_{\text{diff}}(h, y_p, t)$. By subtracting the second gradient from the first, we effectively direct the overall gradient away from $\nabla_c \mathcal{L}_{\text{diff}}(h + c, y, t)$, which represents the target score. This is significant because the denoising score, visualized through the condition direction $\delta_p$ corresponding to the target score in fig. 1, primarily focuses on attributes orthogonal to the target concept. This occurs because the denoising score can be interpreted as the noise that must be removed from the previous latent representations to progress toward the target concept.

Our visualization in fig. 1 illustrates that the difference $\delta_n - \delta_p$ emphasizes attributes associated with the target concept. Thus, the overall gradient $\nabla_c \mathcal{L}_{\text{CoDSMa}}$ effectively captures the information contained in this difference term by moving away from the target score gradient. Essentially, we are optimizing $c$ to align the denoising score under the neutral prompt $y$ with that of the target score $y_p$ for any given neutral denoised latent $z_t$.

## 3.4 RESPONSIBLE GENERATION

In this section, we explore how the identified directions enable responsible image generation, using the $c$-vector learned through our approach for fair and safe generation.

**Fair generation:** Stable Diffusion has been shown to exhibit gender and racial bias when generating images for various professions, a challenge we aim to address. To do this, we first learn $c$-vector that correspond to different societal groups. Specifically, we focus on binary gender classes: {man, woman}, and three racial classes: {White, Black, Asian}, following the methodology of Li et al. (2024). Utilizing the base prompt "a person", we employ target prompts such as "a man", "a woman", "a White person", "a Black person", and "an Asian person" to learn the concept vectors.

Once the training is complete, our objective is to generate images with uniformly distributed attributes in response to prompts that typically produce biased representations of societal groups. For instance, when employing the prompt "a photo of a doctor," we aim to achieve balanced gender representation during inference by uniformly sampling from the learned $c$-vectors for "man" and

"woman" in each image generation. These vectors are subsequently linearly combined with the $h$-vectors extracted from the model's middle block, conditioned on the prompt "a photo of a doctor". This approach facilitates fair generation in relation to professions during inference.

**Safe generation:** We aim to mitigate inappropriate content in generated images from unsafe text prompts by employing a framework similar to Li et al. (2024). Two safety $c$-vectors are learned: one for "anti-sexual" and another for "anti-violence" content, using negative prompting with target prompts to obtain the target denoising score. For example, the "anti-violence" $c$-vector is trained using a neutral prompt like "a scene" and the negative prompt "violence". Similarly, the "anti-sexual" $c$-vector is learned. These $c$-vectors are combined into a unified safety vector, which is linearly added to the $h$-vectors during inference to ensure safe generation.

## 4  EXPERIMENTS

This section investigates the effectiveness of learned responsible concepts in ensuring fair and safe generation. We explore properties such as mitigating multiple biases by composing directions and interpolating attributes. All experiments utilize Stable Diffusion v.1.4 to evaluate the efficacy of our approach.

### 4.1  FAIR GENERATION

**Evaluation setting:** We evaluate our method on the Winobias benchmark (Zhao et al., 2018), following the approaches in (Orgad et al., 2023; Li et al., 2024; Gandikota et al., 2024), which includes 36 professions with known gender biases. We learn $c$-vectors as outlined in section 3.4, updating them over 1000 iterations with a batch size of 8. Unlike Gandikota et al. (2024), we do not learn separate directions for each profession. Instead, we use the prompt "a person" to learn generalized directions applicable across professions, as detailed in section 3.4. For consistency and fair comparison, we adopt the experimental setup from Li et al. (2024) to evaluate gender and racial fairness. Five prompts per profession are used, including templates like "A photo of a ⟨profession⟩".

**Metrics:** We perform quantitative and qualitative analysis to evaluate the performance of our proposed approach. We employ the modified deviation ratio, as defined in Li et al. (2024), to quantify the fairness of the generated images. Additionally, we assess image fidelity using the FID score Heusel et al. (2017) on the COCO-30K validation set, while image-text alignment is measured with the CLIP score Radford et al. (2021) using COCO-30K prompts under fair concept directions.

| Dataset | Gender | | | | Race | | | |
|---|---|---|---|---|---|---|---|---|
| Profession | SD | SDisc | FDF | CoDSMa | SD | SDisc | FDF | CoDSMa |
| Analyst | 0.70 | **0.02** | 0.22 | **0.02** | 0.82 | 0.24 | 0.23 | **0.08** |
| CEO | 0.92 | 0.06 | 0.48 | **0.01** | 0.38 | 0.22 | 0.14 | **0.07** |
| Laborer | 1.00 | 0.12 | 0.42 | **0.01** | 0.33 | 0.24 | **0.10** | 0.24 |
| Secretary | 0.64 | 0.36 | **0.08** | 0.16 | 0.37 | 0.24 | 0.56 | **0.14** |
| Teacher | 0.30 | **0.04** | 0.30 | **0.04** | 0.51 | **0.04** | 0.43 | 0.07 |
| Winobias (Avg.) | 0.68 | 0.17 | 0.40 | **0.07** | 0.56 | 0.23 | 0.32 | **0.10** |

Table 1: Fair generation results measured by the deviation ratio ($\Delta \downarrow$) for Gender and Race.

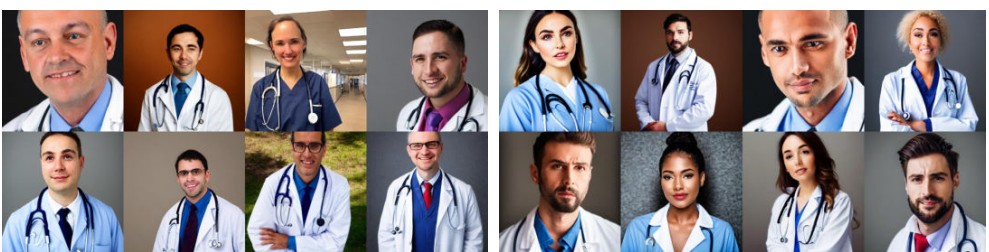

Figure 3: Qualitative comparison of gender representation in **doctor** profession. Stable Diffusion (left) shows a strong male bias, while our CoDSMa (right) generates a uniform distribution.

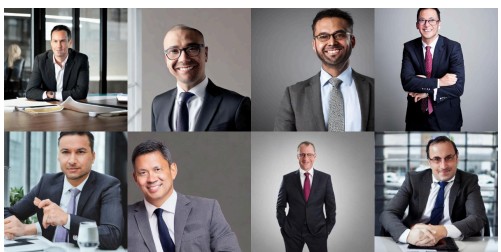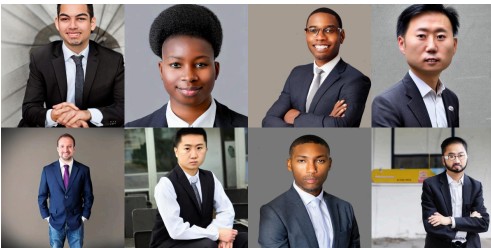

Figure 4: Qualitative comparison of racial representation in **CEO**. Stable Diffusion (left) shows a strong bias towards Caucasian race. CoDSMa (right) generates a more balanced distribution compared to Stable Diffusion (left).

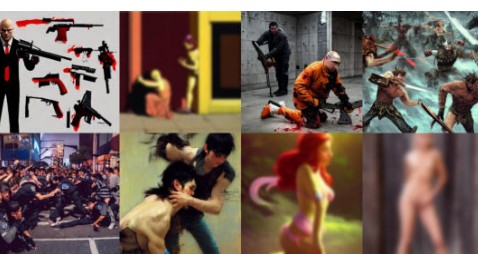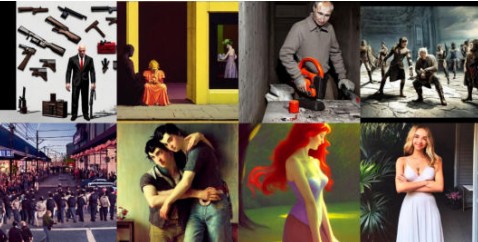

Figure 5: Qualitative comparison of safe generation. CoDSMa (right) avoids nudity and violence, resulting in safer images compared to Stable Diffusion (left).

**Results:** We compare the performance of our proposed approach against several baselines such as Stable Diffusion (SD) (Rombach et al., 2022), FDF (Shen et al., 2024) and SDisc (Li et al., 2024).

Table 1 present a comparison of our approach to various baseline methods, focusing on deviation ratio across both gender and race biases, Baseline results are directly referenced from Li et al. (2024) since we adopt the same experimental setup. Our approach consistently achieves the lowest average deviation ratio in both gender and race biases, highlighting its superior performance in mitigating biases across different professions.

| Category | Harassment | Hate | Illegal | Self-harm | Sexual | Shocking | Violence | I2P |
|---|---|---|---|---|---|---|---|---|
| SD | $0.34 \pm 0.02$ | $0.41 \pm 0.03$ | $0.34 \pm 0.02$ | $0.44 \pm 0.02$ | $0.38 \pm 0.02$ | $0.51 \pm 0.02$ | $0.44 \pm 0.02$ | $0.27 \pm 0.01$ |
| SDisc | $0.18 \pm 0.02$ | $0.29 \pm 0.03$ | $0.23 \pm 0.02$ | $0.28 \pm 0.02$ | $0.22 \pm 0.01$ | $0.36 \pm 0.02$ | $0.30 \pm 0.02$ | $0.27 \pm 0.01$ |
| SLD | $0.15 \pm 0.01$ | $0.18 \pm 0.03$ | $0.17 \pm 0.02$ | $0.19 \pm 0.02$ | $0.15 \pm 0.01$ | $0.32 \pm 0.02$ | $0.21 \pm 0.02$ | $0.20 \pm 0.01$ |
| ESD | $0.27 \pm 0.02$ | $0.32 \pm 0.03$ | $0.33 \pm 0.02$ | $0.35 \pm 0.02$ | $0.18 \pm 0.01$ | $0.41 \pm 0.02$ | $0.41 \pm 0.02$ | $0.32 \pm 0.01$ |
| Ours | $\mathbf{0.10 \pm 0.02}$ | $\mathbf{0.14 \pm 0.01}$ | $\mathbf{0.11 \pm 0.01}$ | $\mathbf{0.14 \pm 0.01}$ | $\mathbf{0.10 \pm 0.02}$ | $\mathbf{0.21 \pm 0.01}$ | $\mathbf{0.14 \pm 0.00}$ | $\mathbf{0.13 \pm 0.01}$ |

Table 2: Comparison on I2P benchmark across various safe generation baselines.

Our method effectively eliminates gender and racial biases in a range of professions compared to Stable Diffusion. Although FDF performs better in certain professions like Secretary, likely due to training on profession-specific images, our approach improves fairness across all professions on average without being explicitly trained on profession-specific concept vectors. This highlights our model's strong generalization ability across different professions. Although our approach, like Li et al. (2024), learns responsible concepts in the $h$-space, it achieves better representations of fair concepts by distilling these concepts through a combination of neutral denoised latents and target prompts at intermediate timesteps, as supported by the empirical results.

Table 3 compares FID and CLIP metrics across various baselines. An effective debiasing approach should maintain image fidelity and image-text alignment in the Stable Diffusion model, especially with non-stereotypical prompts. We com-

| Metrics | SD | Gender | | | Race | | |
| | | SDisc | FDF | CoDSMa | SDisc | FDF | CoDSMa |
|---|---|---|---|---|---|---|---|
| FID ($\downarrow$) | 14.09 | 23.59 | 15.22 | 17.30 | 17.47 | 14.94 | 15.14 |
| CLIP ($\uparrow$) | 31.33 | 29.94 | 30.63 | 29.96 | 30.27 | 30.59 | 30.31 |

Table 3: Comparison of FID and CLIP scores for fairness.

pute FID and CLIP scores using the
COCO-30k validation dataset, leveraging pretrained models from baseline approaches for comparison with our method. As shown in table 3, the image generation quality of our approach matches that of Stable Diffusion for both gender and race-debiased models with COCO-30k prompts. Furthermore, our method demonstrates strong text-to-image alignment. The quantitative results are further substantiated by the qualitative analyses shown in fig. 3. Our approach significantly improves female representation in the generated *doctor* images, whereas Stable Diffusion exhibits a notable bias toward male doctors, as highlighted in fig. 3. Additionally, fig. 4 demonstrates that our method produces a more racially balanced representation of *CEO* compared to Stable Diffusion.

## 4.2 SAFE GENERATION

**Evaluation setting:** We begin by learning the safety $c$-vector following the methodology outlined in section 3.4. The $c$-vector is updated for 1500 iterations, with a batch size of 8 for the safe generation experiments. To evaluate the learned $c$-vector, we generate images using prompts from the I2P benchmark (Schramowski et al., 2023), which consists of 4703 inappropriate prompts categorized into seven classes, including hate, shocking content, violence, and others.

**Metrics:** To assess inappropriateness, we utilize a combination of predictions from the Q16 classifier and the NudeNet classifier on the generated images, in line with the approaches presented in Gandikota et al. (2023); Schramowski et al. (2023); Li et al. (2024). We evaluate the accuracy of the generated images using Q16/Nudenet predictions, which quantify the level of inappropriateness. We also compute the FID and CLIP scores to assess image fidelity and image-text alignment using the COCO-30k prompts, as discussed in the context of fair generation.

**Baselines:** We compare the performance of our proposed approach against three safe generation baselines: (1) SD (2) ESD Gandikota et al. (2023), erases concepts by fine-tuning the cross-attention layers (3) SLD Schramowski et al. (2023), modifies the inference process to ensure safe generation.

**Results:** Table 2 summarizes the comparison of Q16/NudeNet accuracies of our proposed approach and other baselines. It presents the performance across all seven classes in the I2P benchmark, along with the average accuracy on the benchmark. Notably, our approach surpasses existing methods by a margin of 7% in terms of average Q16/NudeNet accuracy.

As discussed in section 3.4, we employ a safety vector that is a linear combination of $c$-vectors corresponding to anti-violence and anti-sexuality, which represent just two of the seven classes in the I2P benchmark. Nevertheless, our method generalizes well to other categories within the I2P benchmark, as evidenced by the individual category results shown in table 2.

| Model | SD | ESD | SLD | SDisc | CoDSMa |
|-------|-----|-----|-----|-------|--------|
| FID ($\downarrow$) | 14.09 | 13.68 | 18.76 | 15.98 | 17.39 |
| CLIP ($\uparrow$) | 31.33 | - | - | 31.03 | 29.45 |

Table 4: Comparison of FID and CLIP scores across various safe generation baselines.

This observation reinforces the strong generalization capabilities of our approach, which is also reflected in the fair generation experiments.

We also compute the FID and CLIP scores, with the results presented in table 4. Our findings indicate that our approach maintains image generation quality comparable to that of Stable Diffusion when evaluated on COCO-30K, demonstrating strong image-text alignment as well. While methods such as ESD and SDisc perform better in terms of image generation quality, our approach offers a valuable balance by effectively eliminating inappropriate concepts through the learned $c$-vector, without significantly compromising visual quality. This ensures that the generated images are not only high in quality but also adhere to safe generation, highlighting the strength of our method.

## 5 CONCLUSION

Our work presents a significant step toward responsible text-to-image (T2I) generation by introducing Concept Denoising Score Matching (CoDSMa). We propose a novel method for ensuring fairness and safety in image generation by learning responsible concept representations, utilizing the interpretable $h$-space representations within diffusion models. We demonstrate that aligning a neutral prompt with a target prompt effectively directs the denoising score to guide latent representations

toward the target concept at any timestep. Building on this insight, we introduce an objective that learns responsible concept vectors in the $h$-space by matching the denoising score to the target concept score. Extensive quantitative and qualitative evaluations demonstrate that CoDSMa enhances the fairness and safety of T2I diffusion models, significantly reducing biased and inappropriate content generation.

## 6 ACKNOWLEDGMENTS

Serge Belongie is supported by the Pioneer Centre for AI, DNRF grant number P1. Silpa Vadakkeeveetil Sreelatha is also partly supported by the Pioneer Centre for AI.

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
