# OpenReview forum: "Concept Denoising Score Matching for Responsible Text-to-Image Generation"
_NeurIPS.cc/2024/Workshop/SafeGenAi — SafeGenAi Poster_

### Official Review · Reviewer_wV2D · 2024-10-08
**Even though the theoretical explanations could be slightly improved, the fundamental approach of h-space guidance is very interesting and is further backed up with strong rigorous experimental results.**

**Rating:** 6
**Confidence:** 4

**Review:**

Summary:
The paper introduces a training objective that includes learnable condition c added to the bottleneck layer of the UNet (so-called h-space). These learned parameters c can then be used at inference to guide the model towards certain concepts (for instance, learning the concept of "man"/"woman"). As an example, when generating "An image of a doctor" the model samples c to be "man" 50% of the time and "woman" 50% of the time, therefore obtaining a gender-neutral representation of "A doctor".

Strengths:
 - The introduced CodSMa objective (described by eq. 4) is very sensible and very clear in itself.
 - The analysis of guidance in the h-space is highly interesting, in this regard this work nicely builds upon the existing literature
 - Very extensive results, which furthermore outline the validity of the approach

Weaknesses:
 - The theoretical explanations behind the approach (3.2 and 3.3) are not very clear and are somewhat misleading
 - In particular, I am left with certain questions:
 -  3.2: I believe the link with Classifier-Free Guidance is necessary in this section. Essentially $\delta_n$ and $\delta_p$ are just
           guidance
           fields towards $y$ and $y_p$. From this perspective, the fact that $\delta_p$ guides *away* from $y_p$ is very surprising. Even
           more surprising is that the argument is made that $\delta_n$ does guide towards $y$. Could the authors clarify why there is such
           a big difference between the workings of this score difference when prompted with $y$ and $y_p$?
           It is also not clear to me how this is related to the CoDSMa objective. From what I understand, the score is matched to the one
           prompted with $y_p$, therefore trying to copy it and not avoid it. Making the connection between section 3.2 and the CoDSMa
           objective clearer would definitely give additional value.
 - 3.3: The entire discussion of the gradient is very misleading and slightly beyond the point. The entire approach is summarized
           in the last sentence of the paragraph. The added value of eq. 7 is not clear to me, I think justifying its presence would benefit the
           paper.

Summary of review:
Even though the theoretical explanations could be slightly improved/rewritten, the approach is very interesting and is further backed up
with strong rigorous experimental results.

General question (for the authors):
When would such an additional guidance condition c be sampled, I suppose that in practice if a user asks for an "apple" it would not be
used? If the approach is activated upon detection of sensitive prompts (such as "job-name"), could the prompt itself not simply be
altered ?

Nits:
 - $L_{diff}$ is not explicitly introduced (should be above line 198)
 - I think that the square of the $L_2$ loss is missing in eq. 4 and 5 (otherwise eq.6 and 7. should be adapted)

---

### Official Review · Reviewer_Ha1E · 2024-10-09
**Promising method for controlled text-to-image generation using learnable attributes in latent space**

**Rating:** 8
**Confidence:** 3

**Review:**

The authors propose and demonstrate a method for controlled text-to-image generation via a method they call _Concept Denoising Score Matching_. The approach leverages a learned transformation on the latent representation in the bottleneck of the diffusion model. The evaluation shows promising empirical results on safety and gender/race bias benchmarks.

The work is largely outside the expertise of the reviewer. In terms of normative motivation for their work, the authors may find the following work useful: https://arxiv.org/abs/2402.05070. In this paper the authors discuss different kinds of alignment objectives, and this work would fit into the "steerable" paradigm.

---

### Official Review · Reviewer_5erL · 2024-10-09
**Insightful proposal but questionable**

**Rating:** 4
**Confidence:** 3

**Review:**

The paper presents an approach to learn responsible concept representations in the feature activation.

The idea is insightful but the design of loss is not convincing to me theoretically. The main concern is whether the sensitive attributes (various “targets”) will end up very close to each other. For example, while “a person” is denoised to be close to “a man” and “a woman”, how about the difference between “a man” and “a woman”? The evaluation does not show the faithfulness for “identity-specific” prompts.
Related to this, it is not clear what is the signal to stop training with this loss. Sharing loss curves will be helpful. Another curve interesting to share is the distance between the targets.
Moreover, maybe some regularization to keep the distance between targets will be worth considering.

Overall: insightful proposal but questionable